

# *Tuber melanosporum* shapes *nir*S-type denitrifying and ammonia-oxidizing bacterial communities in *Carya illinoinensis* ectomycorrhizosphere soils

Zongjing Kang[1,2,*], Jie Zou[1,2,*], Yue Huang[1,2], Xiaoping Zhang[1,2], Lei Ye[1], Bo Zhang[1], Xiaoping Zhang[2] and Xiaolin Li[1]

[1] Soil and Fertilizer Institute, Sichuan Academy of Agricultural Sciences, Chengdu, China
[2] Department of Microbiology, College of Resources, Sichuan Agricultural University, Chengdu, China
[*] These authors contributed equally to this work.

Corresponding author
Xiaolin Li, kerrylee_tw@sina.com

## ABSTRACT

**Background**. *NirS*-type denitrifying bacteria and ammonia-oxidizing bacteria (AOB) play a key role in the soil nitrogen cycle, which may affect the growth and development of underground truffles. We aimed to investigate *nirS*-type denitrifying bacterial and AOB community structures in the rhizosphere soils of *Carya illinoinensis* seedlings inoculated with the black truffle (*Tuber melanosporum*) during the early symbiotic stage.

**Methods**. The *C. illinoinensis* seedlings inoculated with or without *T. melanosporum* were cultivated in a greenhouse for six months. Next-generation sequencing (NGS) technology was used to analyze *nirS*-type denitrifying bacterial and AOB community structures in the rhizosphere soils of these seedlings. Additionally, the soil properties were determined.

**Results**. The results indicated that the abundance and diversity of AOB were significantly reduced due to the inoculation of *T. melanosporum*, while these of *nirS*-type denitrifying bacteria increased significantly. Proteobacteria were the dominant bacterial groups, and *Rhodanobacter*, *Pseudomonas*, *Nitrosospira* and *Nitrosomonas* were the dominant classified bacterial genera in all the soil samples. *Pseudomonas* was the most abundant classified *nirS*-type denitrifying bacterial genus in ectomycorrhizosphere soils whose relative abundance could significantly increase after *T. melanosporum* inoculation. A large number of unclassified *nirS*-type denitrifying bacteria and AOB were observed. Moreover, *T. melanosporum* inoculation had little effect on the pH, total nitrogen (TN), nitrate-nitrogen ($NO_3^-$-N) and ammonium-nitrogen ($NH_4^+$-N) contents in ectomycorrhizosphere soils. Overall, our results showed that *nirS*-type denitrifying bacterial and AOB communities in *C. illinoinensis* rhizosphere soils were significantly affected by *T. melanosporum* on the initial stage of ectomycorrhizal symbiosis, without obvious variation of soil N contents.

## INTRODUCTION

*Tuber* (Ascomycota, Pezizales), commonly known as the truffle, is an edible fungus which can form symbiotic relationships with plants. To date, more than 200 species of truffles have been discovered worldwide (*Bonito et al., 2011a*), particularly in Sweden, France, Italy, Hungary, Spain, the United Kingdom, and Russia (*Wedén, Chevalier & Danell, 2004*). *Tuber melanosporum*, native in France, Italy and Spain, is a valuable and rare species of *Tuber* in the market (*Donnini et al., 2014*). *T. melanosporum* attracted people primarily by its unique aroma compounds. As ectomycorrhizal fungi, truffles need to form symbiotic association with the host plant to complete their life cycle (*Kües & Martin, 2011*). Recently, the cultivation of *T. melanosporum* (as well as other species including *T. aestivum* Vittad., *T. borchii* Vittad., and *T. indicum*) has been conducted worldwide by planting truffle-mycorrhized plant hosts in nurseries (*Reyna & Garcia-Barreda, 2014*), yet studies on the cultivation of this black truffle species in China are seldom reported.

Truffles can form symbiotic relationships with trees from several genera, including *Corylus*, *Quercus*, *Abies Pinus*, *Populus* and *Salix* (*Healy et al., 2016*; *Wan et al., 2016*). The pecan tree, *Carya illinoinensis* (Wangenh.) K. Koch, is an indigenous and economically important tree that grows naturally in moist bottom-land habitats in the United States. However though endemic to the Mississippi basin, *C. illinoinensis* is now cultivated globally (*Wakeling et al., 2001*; *Ruan, Wood & Payne, 1992*). *Tuber lyonii* Butters was the first truffle species described in symbiotic association with the pecan tree (*Trappa, Jumpponen & Cázares, 1996*). *Bonito et al. (2011b)* hypothesized that pecan orchards could be managed to optimize both truffle and pecan production simultaneously. In general, the utilization of *C. illinoinensis* as a host plant for *T. melanosporum* has good application prospects and is worthy of study.

Soil microorganisms and physicochemical properties in the rhizosphere of host plant can vary with the growth of truffles, from the occurrence of ectomycorrhizae to the formation of their mature fruiting bodies (*Ponce et al., 2014*; *Barbieri et al., 2007*; *Garcia-Barreda et al., 2017*). The microbial communities of ectomycorrhizosphere soils are verified to be different from that of rhizosphere soils in both artificial conditions and the wild (*Fu et al., 2016*; *Li et al., 2018a*; *Li et al., 2018b*). There were numerous classified and unclassified microorganisms detected in the ectomycorrhizosphere soils of *Tuber* (*Garcia-Barreda & Reyna, 2012*), which made it difficult to analyze the taxonomy and function of all the bacteria simultaneously. *Paracoccus*, *Pseudomonas*, *Rhizobium*, *Bacillus*, and *Nitrosomonadales* were some possible denitrifying bacteria and ammonia-oxidizing bacteria (AOB) (*Yoshida et al., 2010*; *Purkhold, Pommerening-Röser & Juretschko, 2000*) already detected in the ectomycorrhizae and ectomycorrhizosphere soil of *Tuber*-host associations (*Yang et al., 2019*; *Li et al., 2019*; *Mello et al., 2013*). Nevertheless, the distribution of denitrifying bacteria and the structure of AOB communities in ectomycorrhizosphere soils stay unclear.

The nitrogen (N) cycle is one of the most important nutrient cycles in an ecosystem (*Thamdrup, 2012*), which is vital for host plants and *Tuber*. Both denitrification and ammonia oxidation are important biological processes in the N cycle. Denitrification is the main pathway of $N_2O$ release in farmland ecosystems (*Conrad, 1997*), and an

important process that reduces N pools in farmland soils (*Seitzinger et al., 2006*). By denitrification, denitrifying microorganisms gradually reduce the nitrates or nitrous acid in the environment to gaseous products (NO, $N_2O$, and $N_2$) under multi-step enzymatic catalysis. Most soil denitrification is performed by denitrifying microbes, including bacteria, fungi, and archaea (*Shapleigh, 2006*). Denitrification-related enzymes containing nitrate, nitrite, NO, and $N_2O$ reductase, can be expressed by certain denitrifying bacteria such as *Pseudomonas denitrificans* and *Bacillus stearothermophilus* (*Gregory et al., 2003*). The nitrite reductase functional gene, which controls the catalyzing of the reduction of nitrite to nitric oxide, is the main molecular marker in denitrifying bacteria research (*Braker et al., 2000*). Nitrite reductase has two different structural forms, one of them contains copper (Cu-*nir*) and is encoded by the *nirK* gene, while the other contains the heme CD1 (cd1-*nir*) is encoded by *nirS* gene containing (*Zumft, 1997*). More *nirS* gene are detected than *nirK* gene in most soil samples, and *nirS*-type denitrifying bacteria species in the environment are affected by factors such as water, plants, etc (*Azziz et al., 2017*). As the main driver of the ammonium oxidation process, AOB has long been a focal and functional focus in environmental microbial ecology (*Kowalchuk & Stephen, 2001*). The AOB in $\beta$-Proteobacteria and $\gamma$-Proteobacteria has been considered to be the most active in ammonium oxidation in soils. The oxidation of ammonia to nitrite is a rate-limiting step of nitrification controlled by many AOB ammonia monooxygenase (*amoA*) genes. These genes were widely involved in the study of the relative abundance of AOB in ecosystems (*Purkhold, Pommerening-Röser & Juretschko, 2000*; *Wang et al., 2011*). The efficiency of nitrification and denitrification in soils was related to the number of microbial functional genes. The community structure of ammonia-oxidizing microorganisms, however, could in return be affected by the concentration of ammonium nitrogen and organic matter (*Petersen et al., 2012*). Structures of *nirS*-type denitrifying bacterial and AOB community in *Tuber* ectomycorrhizosphere soils are unclear. Only few studies have focused on the relationship between these bacteria and properties of the ectomycorrhizosphere soils.

In our study, we therefore infected *C. illinoinensis* with the spores of *T. melanosporum* and tracked the formation of mycorrhizae during the next six months of cultivation. We also did next-generation amplicon sequencing for *nirS* and *amoA* genes, to analyze the denitrification and ammonia oxidation microbial community structures in rhizosphere soils. Furthermore, the physicochemical soil properties in the rhizosphere of *C. illinoinensis* seedlings were determined. The study aims to investigate *nirS*-type denitrifying bacterial and AOB community structures involving in the ammonium oxidation and denitrification processes in the *T. melanosporum* ectomycorrhizosphere soils.

## MATERIALS & METHODS

### Cultivation of sterile *C. illinoinensis* seedlings

Seeds of *C. illinoinensis* were soaked in fresh-water for about 20 h and sterilized with 0.3% potassium permanganate for 30 min, then the seeds were rinsed with distilled water until the last wash solution became colorless. In order to increase the germination rate, the seeds were stored in the sands which had been sterilized for 90 min at 121 °C in an autoclave.

One month later, the germinated *C. illinoinensis* seeds were sown in a plastic container filled with the sterilized substrate (vermiculite, perlite and water at a ratio of 1:1:1, v/v/v) (Fig. S1) (*Li et al., 2017*). The plastic container with seeds was then placed in a natural light plastic greenhouse and watered with sterile water, to maintain a soil moisture content of 25%. The temperature of the greenhouse was 23 °C–25 °C in daytime and 16 °C–20 °C at night. The well-grown *C. illinoinensis* seedlings with similar plant height and stem thickness were selected for inoculation.

## Truffle inoculation treatment

*T. melanosporum* ascocarps were purchased from truffle producing areas in French and used as spore inoculum. The ascocarps were washed with sterile water and their surface sterilized by burning briefly with 75% alcohol. Then the ascocarps were soaked in sterile water to stimulate spores release and germination (*Semeniuk, 2008*). Finally, the treated *T. melanosporum* ascocarps were smashed and blended using food chopper to obtain their paste. Next, the substrates with peat, vermiculite, perlite organic soil and water at a ratio of 1:1:1:0.9 (v/v/v/v) were prepared and sterilized for 90 min at 121 °C in an autoclave. The sterilized substrate (total nitrogen 2.05 g/kg, nitrate-nitrogen 37.50 mg/kg and ammonium-nitrogen 20.16 mg/kg) was filled into nursery pots and 1.5 g of the paste inoculum was mixed into the substrate of each pot, then one sterile seedling was transplanted into each nursery pot. Thirty seedlings inoculated with *T. melanosporum* were named group I. At the same time, an equivalent number of *C. illinoinensis* seedlings were transplanted into nursery pots without *T. melanosporum* inoculum, named group II. Each group had thirty replicates to ensure the survival of enough seedlings after 6 months cultivation. All pots were maintained in a greenhouse under the same conditions without fertilization, and they were watered with tap water every three days at 6:00 pm for 6 month (*Geng et al., 2009*).

## Sampling strategy and soil analyses

Eight pots of group I and four pots of group II were randomly selected, and roots of these seedlings were observed under a microscope. Morphological and anatomical analysis revealed that the four seedlings in group II were not colonized by ectomycorrhizal fungi, and only 4 of the 8 seedlings in group I inoculated with truffle spores, showing ectomycrrhizae formation (Fig. 1) (*Marozzi et al., 2017*). The four pots of group I colonized by *T. melanosporum* were used as experimental materials, and the four pots of group II were used as controls, thus, each experimental treatment had four replicates. Ectomycorrhizosphere soils (rhizosphere soils of *C. illinoinensis* seedlings inoculated with *T. melanosporum*) and rhizosphere soils (rhizosphere soils of *C. illinoinensis* seedlings without inoculating) samples were sampled and placed into 2 mL EP tubes. The samples were cooled rapidly using liquid nitrogen and stored at −80 °C. The remaining soils around the *C. illinoinensis* roots were air dried for soil property analysis, including pH, organic matter (OM), available potassium (AK), total nitrogen (TN), available phosphorus (AP), nitrate-nitrogen ($NO_3^-$-N) and ammonium-nitrogen ($NH_4^+$-N), which were determined following the methods described in previous studies (*Lu, 1999*; *Li et al., 2017*). Ectomycorrhizosphere soils of *C. illinoinensis*

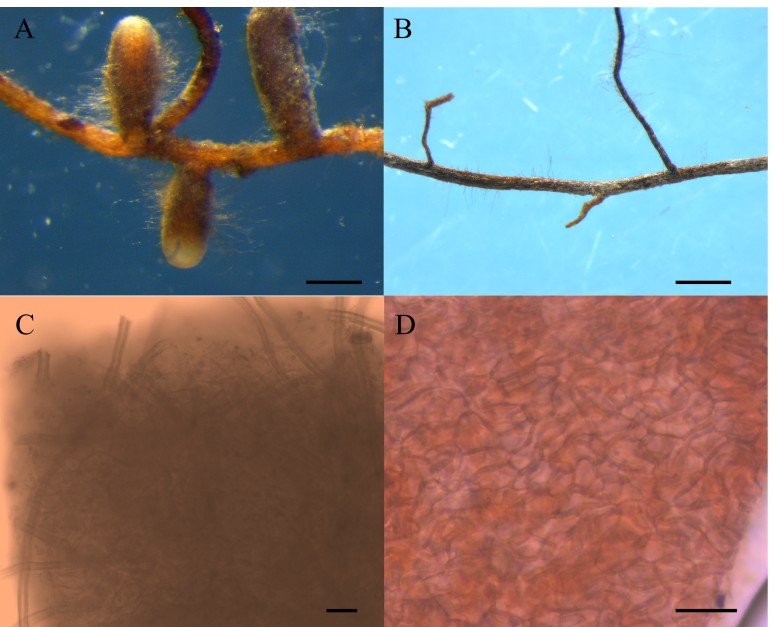

**Figure 1** **Morphological features of *Carya illinoinensis* root in association with or without *Tuber melanosporum*.** (A) Ectomycorrhizae of *Tuber melanosporum*, *Scale bar* = 500 μm; (B) Root of *Carya illinoinensis* without *Tuber melanosporum*, *Scale bar* = 700 μm; (C) Mantle with cystidia , *Scale bar* = 65 μm; (D) Mantle cell of ectomycorrhizae, *Scale bar* = 10 μm.

colonized with *T. melanosporum* were labeled as "FH" from Chines name of the *Tuber*, "Faguo Heibao", and the rhizosphere soils of the *C. illinoinensis* seedlings without *T. melanosporum* inoculation were labeled as "CK" standing for Control Check.

## DNA extraction, PCR amplification, and MiSeq sequencing

DNA from rhizosphere soils was extracted using a Soil DNA Kit (D5625-01, Omega Bio-tek Inc., Norcross, GA, USA) following the manufacturer's instructions. The DNA concentration and purity were measured on 1% agarose gels, adjusted to 1 ng/μL using sterile water and stored at −20 °C.

The universal primers Cd3aF (5′-GTSAACGTSAAGGARACSGG-3′) (*Michotey, Méjean & Bonin, 2000*) and R3cd (5′-GASTTCGGRTGSGTCTTGA-3′) (*Throbäck et al., 2004*) were used to amplify *nirS*-type denitrifying bacteria in PCR. The reaction consisted of 10 ng DNA template, 4 μL FastPfu buffer, 2 μL dNTPs (2.5 mmol L$^{-1}$), 0.4 μL FastPfu polymerase, 0.8 μL forward primer (5 μmol L$^{-1}$), and 0.8 μL reverse primer (5 μmol L$^{-1}$), diluted to 20 μL with ddH$_2$O. The reaction program was set as follows: pre-denaturation at 95 °C for 4 min, followed by 28 cycles of denaturation at 95 °C for 40 s, annealing at 56 °C for 30 s, extension at 72 °C for 40 s, and then final extension at 72 °C for 10 min. The universal primers b-amoA-1F (5′-GGGGTTTCTACTGGTGGT-3′) and b-amoA-2R (5′-CCCCTCKGSAAAGCCTTCTTC-3′) (*Mao, Yannarell & Mackie, 2011*) were used to amplify the AOB in a 30 μL reaction system, which were consisted of 15 μL 2 × Ex Taq MasterMix, 1.2 μL forward and reverse primers (10 μmol/L), 3 μL DNA template and

9.6 µL RNase-free water. Samples were amplified by subjecting them to the following PCR program: 95 °C for 3 min followed by 40 cycles at 95 °C for 60s, 55 °C for 45 s and 72 °C extension for 60 s. The samples were then mixed and analyzed by 1.5% agarose gel electrophoresis, after which the PCR product was recovered using a gel recovery kit (Axygen, USA) and eluted with Tris–HCl and 1.5% agarose gel electrophoresis (*Kim et al., 2011*). According to the preliminary quantitative electrophoresis results, PCR products were quantified using QuantiFluor$^{TM}$-ST Blue Fluorescence Quantitation System (Promega, USA), after which the proportions were mixed according to template concentration for each sample. Finally, these samples stored at −80 °C.

High throughput sequencing was conducted at Shanghai Majorbio Bio-pharm Technology Co., Ltd. and sequenced according to manufacturer's recommendations with the Illumina MiSeq sequencing platform. Raw data were submitted to the Sequence Read Archive (SRA) database with the accession number SRR10522644–SRR10522659.

## High-throughput data analysis and statistical analysis

Paired-end reads were obtained, and paired-end reads that overlapped were merged using FLASH (Version 1.2.11) (*Tanja & Salzberg, 2011*). The high-quality sequences with mismatch rate of overlap sequence lower than 20% and with barcode and primer sequence were filtered by QIIME (Version 1.8.0) and Trimmomatic software (*Caporaso et al., 2010*), then correct sequence direction. Uparse (Version 7.0.1090) software was used to analyze the high-quality sequences, and these sequences with ≥97% similarity were clustered to the same OTU (Operational Taxonomic Units) (*Edgar, 2013*). A representative sequence for each OTU was screened for further annotation, and the taxonomic analysis of OTUs was performed using the RDP Classifier (Version 2.11; http://sourceforge.net/projects/rdp-classifier/) (*Wang et al., 2007*) based on the FunGene (Release7.3; http://fungene.cme.msu.edu/) database. Mothur (Version 1.30.2) was used to divided OTUs, and alpha diversity analysis of *nirS*-type denitrifying bacterial and AOB population (Coverage, Chao1, ACE, Shannon and Simpson index) were performed based on these OTUs. Source code for the OTUs data is available at: https://github.com/catherinekang/bacterial-community.git. Moreover, a rarefaction curve was used to evaluate the representativeness of sequencing amount for the diversity of the original *nirS*-type denitrifying bacteria or the AOB. In addition, barplots based on the R software (Version 3.1.3) (*Core et al., 2011*) were utilized to cluster and analyze the phylum and genus to share the taxonomic composition of the soil microbial communities. Principle component analysis (PCA) and clustering analysis were used to reflect the beta diversity.

Alpha diversity estimators, soil properties and relative abundance of the taxa of AOB and *nirS*-type denitrifying bacteria between the two treatments were compared with independent *t-test*, which was performed in SPSS v21.0 (IBM Inc., Armonk, NY, USA). All the data were presented as the mean value ± standard deviation (SD) of the four biological replicates in each treatment group. All the significant differences were assessed at $P < 0.05$.

**Table 1  Physical and chemical properties of soil samples.**

| Sample | pH | OM (g/kg) | TN (g/kg) | $NH_4^+$-N (mg/kg) | $NO_3^-$-N (mg/kg) | AK (mg/kg) | AP (mg/kg) |
|--------|-----|-----------|-----------|---------------------|---------------------|-------------|-------------|
| **CK** | $9.48 \pm 0.06$ | $37.23 \pm 1.63$ | $1.08 \pm 0.03$ | $10.80 \pm 0.45$ | $11.36 \pm 0.12$ | $199.52 \pm 31.15$ | $18.20 \pm 0.51$ |
| **FH** | $9.40 \pm 0.02$ | $34.19 \pm 1.23$ | $0.95 \pm 0.04$ | $12.09 \pm 0.07$ | $11.97 \pm 0.04$ | $244.74 \pm 10.09$ | $18.92 \pm 1.49$ |

**Notes.**

OM, organic matter; TN, total nitrogen; $NH_4^+$-N, ammonium-nitrogen; $NO_3^-$-N, nitrate-nitrogen; AK, available potassium; AP, available phosphorus; FH, ectomycorrhizosphere soil (the rhizosphere soil of *Carya illinoinensis* mycorrhized with *Tuber melanosporum*); CK, the rhizosphere soil of *Carya illinoinensis* without *Tuber melanosporum* partner.

Values are mean $\pm$ standard deviation ($n = 4$). There is no significant difference between two treatments ($P > 0.05$).

## RESULTS

### Soil property analysis

The results of soil properties showed that the nitrate-nitrogen and available potassium contents of FH were slightly higher than those of CK, and there were no significant different in pH, organic matter, available potassium, total nitrogen, available phosphorus, nitrate-nitrogen and ammonium-nitrogen between CK and FH (Table 1).

### AOB diversity analysis

A total of 123,091 reads were obtained from the 8 samples after quality control procedures, most of which were 401–500 bp, and there were 11,990–19,959 reads in each sample (Fig. 2A). All the sequences were clustered into 251 OTUs, and the two treatments shared 47 OTUs. The number of unique OTUs which could only be detected in FH (77) was smaller than the unique OTUs which could only be detected in CK (127) (Fig. 3A).

As shown in Table 2, the coverage in the two treatments were 99%, indicating that we sequenced ammonia monooxygenase genes at the proper depth and the data were reliable. The richness indices such as ACE ($P = 0.049$) and Chao1 ($P = 0.043$) estimators in CK was significantly greater than those in FH. Shannon diversity in the two treatments ranged from 1.32 to 1.62, and was significantly greater in CK treatment ($P = 0.047$), however, there was no significant difference between FH and CK in the Simpson index and the observed species (OTU). For the observed species, Shannon, Chao1, and ACE of FH were lower than those of CK, which showed that the richness and diversity of AOB in FH was significantly lower than that in CK ($P < 0.05$) (Table 2).

### Taxonomic analyses of AOB

As shown in Fig. 4A, the observed species mainly belonged to two phyla, and Proteobacteria was the most abundant in each treatment. The average relative abundance of Proteobacteria in CK (98.59%) was significantly greater than that in FH (84.93%) ($P = 0.049$) (Table 3).

The results of classification showed that 251 OTUs were separated into more than six genera (Fig. 5A). The primary three genera in FH treatment were *Nitrosospira* (57.90%), *Nitrosomonas* (25.12%) and unclassified bacteria (15.07%); while those in CK were *Nitrosospira* (55.38%), *Nitrosomonas* (39.99%) and an unclassified genus belong to Betaproteobacteria (1.93%) (Table 4). Moreover, the relative abundance of *Nitrosospira* in both treatments was approximately the same, while the relative abundance of *Nitrosomonas* in CK was slightly greater than that in FH. *Nitrosococcus* was detected in CK, with

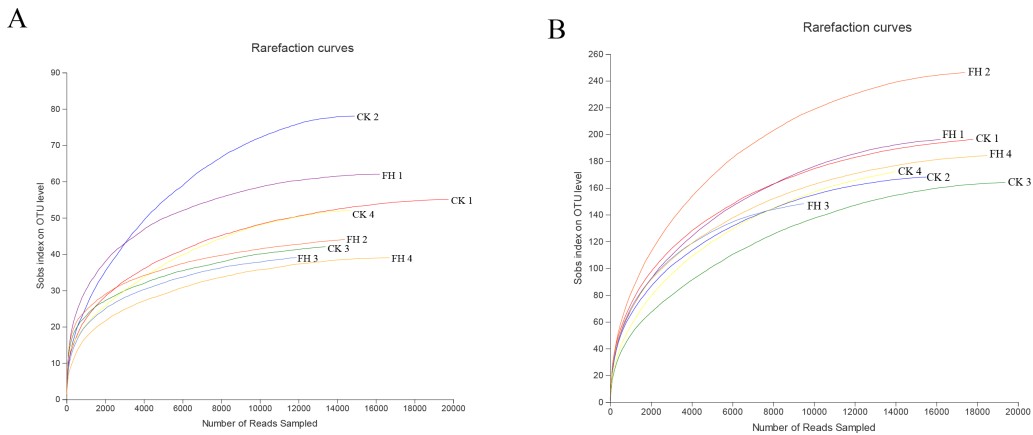

**Figure 2** **Rarefaction curves for ammonia-oxidizing bacterial (A) and *nirS*-type denitrifying bacterial (B) OTU diversity in different samples.** FH, ectomycorrhizosphere soil (the rhizosphere soil of *Carya illinoinensis* mycorrhized with *Tuber melanosporum*). CK, the rhizosphere soil of *Carya illinoinensis* without *Tuber melanosporum* partner.

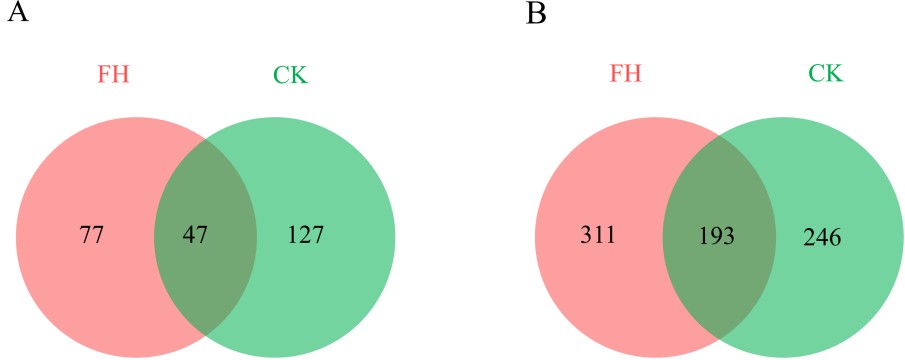

**Figure 3** **Numbers of shared and unique ammonia-oxidizing bacterial (A) and *nirS*-type denitrifying bacterial (B) Operational Taxonomic Units (OTUs).** FH, ectomycorrhizosphere soil (the rhizosphere soil of *Carya illinoinensis* mycorrhized with *Tuber melanosporum*). CK, the rhizosphere soil of *Carya illinoinensis* without *Tuber melanosporum* partner.

significantly higher relative abundance than that in FH ($P = 0.046$). Nevertheless, the most abundant unclassified AOB genus in FH was significantly more abundant than that in CK (1.41%) ($P = 0.005$).

## *NirS*-type denitrifying bacterial diversity analysis

A total of 139,493 reads were obtained from the 8 samples after quality control procedures, most of which the sequencing length was 301–400 bp, and there were 10,424-19,961 reads in each sample (Fig. 2B). All the sequences from the two treatments were clustered into 750 OTUs, which were utilized to form a histogram of genera and phyla. Moreover, there were a total of 193 OTUs shared by two treatments and the number of unique OTUs detected in FH (311) was more than the unique OTUs detected in CK (246) (Fig. 3B).

**Table 2** Community richness and diversity indices of ammonia-oxidizing bacteria and *nirS*-type denitrifying bacteria in rhizosphere soil with or without *Tuber melanosporum* partner.

| Samples | | Observed species | Simpson | Shannon | ACE | Chao1 | Coverage |
|---|---|---|---|---|---|---|---|
| **AOB** | CK | 56.75 ± 15.21 | 0.32 ± 0.04 | 1.62 ± 0.15 | 58.68 ± 2.61 | 57.22 ± 6.63 | 0.99 |
| | FH | 46.00 ± 10.92 | 0.45 ± 0.15 | 1.32 ± 0.19[*] | 48.13 ± 8.17[*] | 46.55 ± 5.09[*] | 0.99 |
| | T-statistic | 1.148 | −1.722 | 2.482 | 2.460 | 2.551 | 0.069 |
| | *P*-value | 0.295 | 0.136 | 0.048 | 0.049 | 0.043 | 0.947 |
| ***nirS*-type denitrifying bacteria** | CK | 175.00 ± 14.38 | 0.16 ± 0.05 | 2.55 ± 0.16 | 188.43 ± 10.19 | 179.61 ± 14.37 | 0.99 |
| | FH | 193.50 ± 40.51 | 0.13 ± 0.03 | 2.80 ± 0.08[*] | 204.27 ± 7.73[*] | 196.39 ± 40.51[*] | 0.99 |
| | T-statistic | −0.861 | 0.928 | −2.675 | −2.584 | −2.475 | −0.047 |
| | *P*-value | 0.422 | 0.389 | 0.037 | 0.042 | 0.048 | 0.964 |

Notes.

FH, ectomycorrhizosphere soil (the rhizosphere soil of *Carya illinoinensis* mycorrhized with *Tuber melanosporum*); CK, the rhizosphere soil of *Carya illinoinensis* without *Tuber melanosporum* partner.

Values are mean ± standard deviation ($n = 4$).

[*]Significant difference between samples ($P < 0.05$).

The coverage of the denitrifying bacterial communities in the two treatments was 99% (Table 2). The ACE ($P = 0.042$) and Chao1 ($P = 0.048$) estimators revealed that the richness of *nirS*-type denitrifying bacteria in FH was significantly greater than that in CK. The Simpson index of two treatments was similar, and the number of observed species of FH was slightly higher than that of CK without significant difference. And the Shannon index of FH was significantly higher than that of CK ($P = 0.037$) (Table 2). The results demonstrated that the richness and diversity of *nirS*-type denitrifying bacteria were significantly increased with *T. melanosporum* inoculation. Moreover, the observed species of *nirS*-type denitrifying bacteria were far more than AOB in both treatments (Fig. S2), the Shannon, Chao1 and ACE index of *nirS*-type denitrifying bacteria were much higher than AOB.

## Taxonomic analyses of *nirS*-type denitrifying bacterial communities

As shown in Fig. 4B, at the phylum level, the most abundant were unclassified bacteria and no rank bacteria, followed by Proteobacteria. The average proportions of Proteobacteria in the two treatments were 9.49% (CK) and 8.69% (FH), with Proteobacteria and an unclassified bacteria phylum being slightly more abundant in CK than in FH (Table 3). Nevertheless, there was no significant difference in the relative abundance of the dominant phyla between FH and CK.

The results showed that these 750 OTUs belonged to more than nine genera, and the most abundant classified genera were *Pseudomonas*, *Rhodanobacter*, *Magnetospirillum* and *Rubrivivax* (Fig. 5B). The main classified genera in FH were *Pseudomonas* (0.46%), while the main classified genus in CK was *Rhodanobacter* (0.33%), and the relative abundance of *Pseudomonas* of FH was significantly higher than that of CK ($P = 0.017$) (Table 4). The relative abundance of the no rank bacterial genus was similar in FH and CK, without significant difference. However, the relative abundance of two unclassified genera observed in FH, belonging to Gammaproteobacteria ($P = 0.027$) and Betaproteobacteria ($P = 0.015$), were significantly lower than those observed in CK.

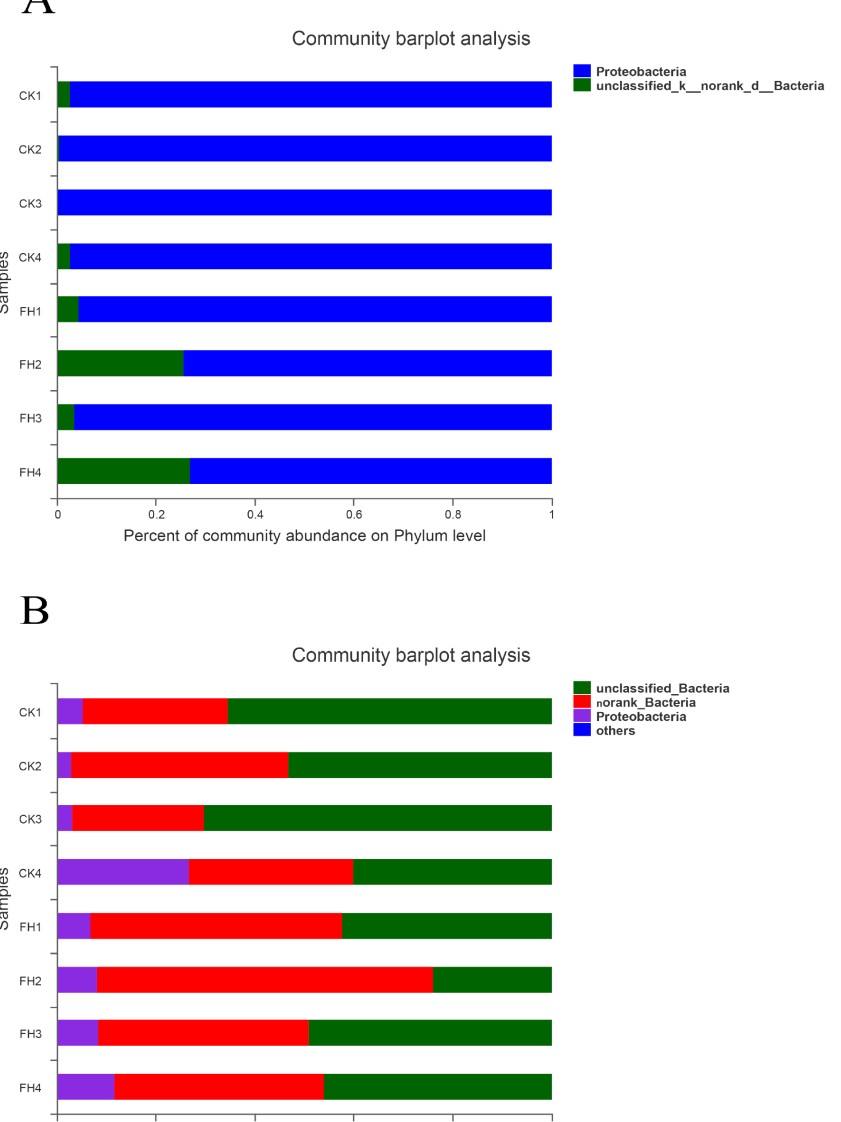

**Figure 4** **Taxonomic composition of ammonia-oxidizing bacterial (A) and *nirS*-type denitrifying bacterial (B) communities at the phylum levels.** FH, ectomycorrhizosphere soil (the rhizosphere soil of *Carya illinoinensis* mycorrhized with *Tuber melanosporum*). CK, the rhizosphere soil of *Carya illinoinensis* without *Tuber melanosporum* partner.

## Principle component analysis (PCA)

The Unifrac-PCA analysis was used to visualize the similarities and differences of the bacterial component in different soil samples. The shorter distance between two samples means more similar *nirS*-type denitrifying bacterial and AOB communities in the different samples, and the different shaded circles represented confidence ellipses. The results showed that both AOB and *nirS*-type denitrifying bacteria in the FH were not similar to CK. In

**Table 3  The relative abundance of the most abundant ammonia-oxidizing and *nirS*-type denitrifying bacterial phyla in rhizosphere soil with or without *Tuber melanosporum* partner.**

|  | Phyla | CK | FH | T-statistic | *P*-value |
|---|---|---|---|---|---|
| **AOB** | Proteobacteria | 98.59% ± 0.01 | 84.93% ± 0.13[*] | 2.467 | 0.049 |
|  | Unclassified Bacteria | 1.41% ± 0.01 | 15.07% ± 0.13[*] | −2.594 | 0.041 |
|  | Proteobacteria | 9.49% ± 0.12 | 8.69% ± 0.02 | 0.135 | 0.897 |
|  | No rank Bacteria | 33.25% ± 0.08 | 51.10% ± 0.12 | −2.154 | 0.074 |
| **nirS-type denitrifying bacteria** | Unclassified Bacteria miscellaneous | 0.04% ± 0.00 | 0.03% ± 0.00 | 0.308 | 0.769 |
|  | Unclassified Bacteria | 57.22% ± 0.14 | 40.18% ± 0.11 | 1.934 | 0.101 |

Notes.

FH, ectomycorrhizosphere soil (the rhizosphere soil of *Carya illinoinensis* mycorrhized with *Tuber melanosporum*); CK, the rhizosphere soil of *Carya illinoinensis* without *Tuber melanosporum* partner.

Values are mean ± standard deviation ($n = 4$).

*Significant difference between samples ($P < 0.05$).

Fig. 6, the distance between FH and CK in AOB was greater than the distance between them in *nirS*-type denitrifying bacteria, which could indicate that the difference of AOB communities between FH and CK was greater than that of *nirS*-type denitrifying bacterial communities between them.

## DISCUSSION

This experiment amplified *nirS* and *amoA* genes partner to detect the denitrifying bacterial and ammonia-oxidizing bacterial (AOB) communities from the rhizosphere soil of *C. illinoinensis* with or without *T. melanosporum* inoculation. The study also explored the impacts of *T. melanosporum* inoculation on the nitrogen cycling bacteria in microecology of *C. illinoinensis* rhizosphere within 6 months after inoculation.

Results of PCA analysis showed a significant difference of *nirS*-type denitrifying bacterial and AOB communities between ectomycorrhizosphere soil and rhizosphere soil. This was in accordance with the previous finding that the composition and function of bacterial communities in ectomycorrhizosphere soil could be altered by the formation of ectomycorrhizae (*Jung et al., 2012*). In our study, the total nitrogen, nitrate-nitrogen and ammonium-nitrogen contents of ectomycorrhizosphere soil were not significantly different from rhizosphere soil similarly to the contents of other determined properties. Previous studies reported that the ectomycorrhizal fungi could increase available nitrogen in forest soil by producing enzymes or organic acids, accelerating plant litter decomposition, and soil mineral dissolution (*Sterkenburg et al., 2018*; *Lindahl et al., 2006*). The contrasting results from our study, however, suggested that the ectomycorrhizae of *T. melanosporum* had little impact on the soil properties in the host plant rhizosphere which did not contain plant litter at the initial stage of inoculation.

Moreover, the richness and diversity of AOB in the ectomycorrhizosphere soil were significantly lower than those in rhizosphere soil ($P < 0.05$). The Proteobacteria was the most abundant AOB in both ectomycorrhizosphere soil (84.93%) and rhizosphere soil (98.59%), which was consistent with the in forest soils (*Rösch, Mergel & Bothe, 2002*). The most abundant genera of AOB were *Nitrosospira* and *Nitrosomonas*. *Li et al. (2018a)*

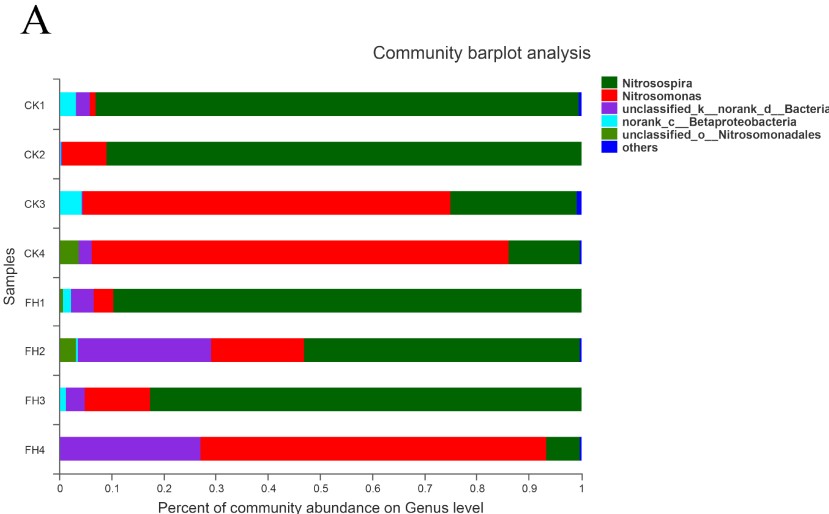

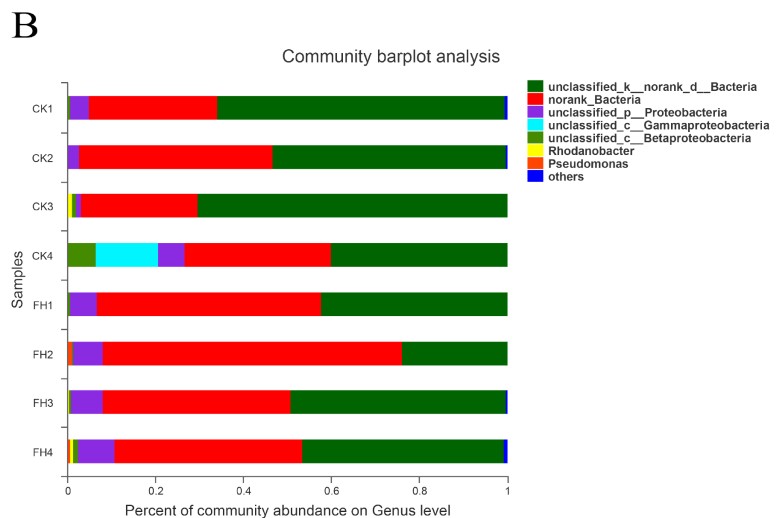

**Figure 5** **Taxonomic composition of ammonia-oxidizing bacterial (A) and *nirS*-type denitrifying bacterial (B) communities at the genus levels.** FH, ectomycorrhizosphere soil (the rhizosphere soil of *Carya illinoinensis* mycorrhized with *Tuber melanosporum*). CK, the rhizosphere soil of *Carya illinoinensis* without *Tuber melanosporum* partner.

and *Li et al. (2018b)* reported that the *Nitrosospira* and *Nitrosomonas* were the most abundant AOB in ectomycorrhizosphere soil of *Pinus massoniana* Lamb colonized by *Pisolithus tinctorius* under field condition, which is basically the same with the findings of our study. In addition, the relative abundance of *Nitrosococcus* and an unclassified AOB genus of Betaproteobacteria in ectomycorrhizosphere soils was significantly greater than that in rhizosphere soils while another unclassified genus was significantly lower than that in rhizosphere soils ($P < 0.05$). The soil moisture might have an indirect effect on AOB communities by changing the availability and mobility of nitrogen in the substrate (*Gleeson et al., 2010*; *Hu et al., 2015*). However, *C. illinoinensis* seedlings in two treatments

**Table 4   The relative abundance of the most abundant ammonia-oxidizing and *nirS*-type denitrifying bacterial genera in rhizosphere soil with or without *Tuber melanosporum* partner.**

|  | Genera | CK | FH | *T*-statistic | *P*-value |
|---|---|---|---|---|---|
| **AOB** | *Nitrosospira* | 55.38% ± 0.42 | 57.90% ± 0.38 | −0.089 | 0.932 |
|  | *Nitrosomonas* | 39.99% ± 0.41 | 25.12% ± 0.28 | 0.600 | 0.573 |
|  | *Nitrosococcus* | 0.12% ± 0.00 | 0.00% ± 0.00[*] | 2.509 | 0.046 |
|  | Unclassified Betaproteobacteria | 1.93% ± 0.02 | 0.85% ± 0.01[*] | 2.533 | 0.044 |
|  | Unclassified Bacteria | 1.41% ± 0.01 | 15.07% ± 0.13[*] | −4.409 | 0.005 |
|  | Unclassified Nitrosomonadales | 0.98% ± 0.02 | 0.97% ± 0.01 | 0.011 | 0.991 |
| **nirS-type denitrifying bacteria** | No rank Bacteria | 33.25% ± 0.08 | 51.10% ± 0.12 | −1.940 | 0.102 |
|  | Unclassified Bacteria | 57.22% ± 0.14 | 40.18% ± 0.11 | 1.934 | 0.101 |
|  | Unclassified Proteobacteria | 3.44% ± 0.02 | 7.16% ± 0.01 | −2.268 | 0.095 |
|  | Unclassified Gammaproteobacteria | 3.6% ± 0.07 | 0.00% ± 0.00[*] | 2.910 | 0.027 |
|  | Unclassified Betaproteobacteria | 1.98% ± 0.03 | 0.51% ± 0.00[*] | 5.040 | 0.015 |
|  | *Rhodanobacter* | 0.33% ± 0.01 | 0.35% ± 0.00 | −0.061 | 0.953 |
|  | *Pseudomonas* | 0.03% ± 0.00 | 0.46% ± 0.00[*] | −2.541 | 0.017 |
|  | *Magnetospirillum* | 0.01% ± 0.00 | 0.01% ± 0.00 | −0.049 | 0.963 |
|  | *Rubrivivax* | 0.01% ± 0.00 | 0.04% ± 0.00 | −0.981 | 0.396 |

**Notes.**

FH, ectomycorrhizosphere soil (the rhizosphere soil of *Carya illinoinensis* mycorrhized with *Tuber melanosporum*); CK, the rhizosphere soil of *Carya illinoinensis* without *Tuber melanosporum* partner.

Values are mean ± standard deviation ($n = 4$).

[*]Significant difference between samples ($P < 0.05$).

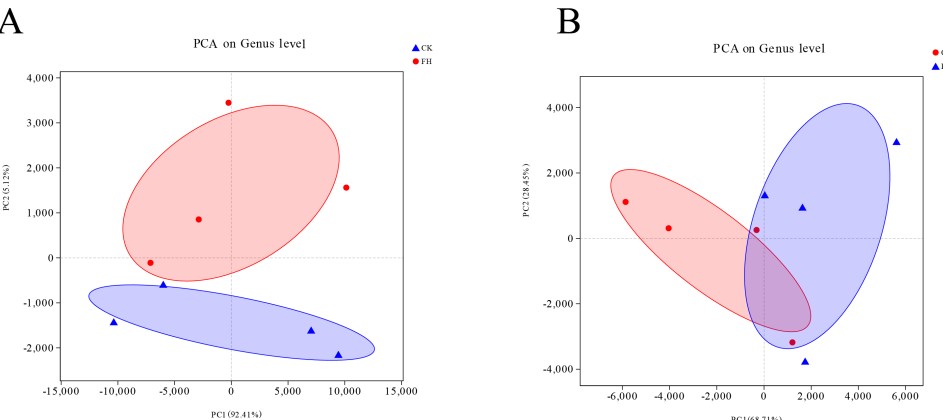

**Figure 6   Principal component analysis (PCA) of ammonia-oxidizing bacterial (A) and *nirS*-type denitrifying bacterial (B) communities.** FH, ectomycorrhizosphere soil (the rhizosphere soil of *Carya illinoinensis* mycorrhized with *Tuber melanosporum*). CK, the rhizosphere soil of *Carya illinoinensis* without *Tuber melanosporum* partner.

were watered equally in this work. Previous studies have illustrated that the diversity of bacterial communities reduced in the brûlés (an area devoid of herbaceous cover) and ectomycorrhizosphere soil (*Mello et al., 2013*; *Zhang et al., 2019*). Nevertheless, some bacterial genera in ectomycorrhizosphere soils were significantly more abundant than those in rhizosphere soils, such as *Mesorhizobium, Reyranena, Rhizomicrobium* and *Nordella* (*Li*
*et al., 2017*). These findings may imply that the *T. melanosporum* ectomycorrhizae have different effect on different AOB genera (*Doornbos, Van Loon & Bakker, 2012*), and the richness and diversity of the AOB in rhizosphere soils were reduced by *T. melanosporum* inoculation in general, though the ectomycorrhizae of *T. melanosporum* indeed affect the relative abundance of *Nitrosospira* and *Nitrosomonas* little during early symbiotic stage.

In terms of the structural analysis *nirS*-type denitrifying bacterial community, Proteobacteria was the dominant group both in rhizosphere and ectomycorrhizosphere soils. Previous studies showed that most of the denitrifying bacteria which were detected in crop rhizosphere soils and arable land soils belong to Proteobacteria (*Chang et al., 2012*; *Hui et al., 2010*; *Hou et al., 2018*), which corroborates our results. Our work also showed that the *Rhodanobacter*, *Pseudomonas*, *Magnetospirillum* and *Rubrivivax* were the dominant classified bacterial genera in all the soil samples, which were detected in plant rhizosphere soils and arable land soils frequently (*Yu et al., 2018*; *Wen et al., 2016*). The richness and diversity of *nirS*-type denitrifying bacterial communities in ectomycorrhizosphere soil were significantly greater than those in rhizosphere soil ($P < 0.05$). The land use and fertilizer regimes affected the biological activity of denitrifying bacteria (*Hui et al., 2010*), and the abundance of *Rhodanobacter* was sensitive to pH (*Green et al., 2010*). However, the differences in soil properties between two treatments were not significant in the present study, which may illustrate that the abundance of some *nirS*-type denitrifying bacterial communities in rhizosphere soils was increased due to *T. melanosporum* inoculation. The previous study showed that the mycorrhiza and rhizobia assist plants with the uptake of phosphorus and nitrogen, respectively (*Van der Heijden, Bardgett & Van Straalen, 2008*), which may suggest that the *T. melanosporum* ectomycorrhizae accelerate the weathering of minerals and the degradation of recalcitrant organic matter by enhancing the colonization of *nirS*-type denitrifying bacteria in ectomycorrhizosphere soils (*Berendsen, Pieterse & Bakker, 2012*; *Deveau et al., 2016*). Meanwhile, the relative abundance of unclassified bacterial genera belonging to Gammaproteobacteria and Betaproteobacteria were significantly lower in ectomycorrhizosphere soils ($P < 0.05$), which demonstrated that the colonization of *T. melanosporum* decreased the relative abundance of some *nirS*-type denitrifying bacterial genera in ectomycorrhizosphere soils.

Our work revealed that the *Pseudomonas* was the most abundant classified *nirS*-type denitrifying bacteria in ectomycorrhizosphere soils, which are widely distributed in plant rhizosphere soils and truffle orchard soils (*Prieme, Braker & Tiedje, 2002*; *Sharma et al., 2005*). Furthermore, *Pseudomonas* were also detected in fruiting bodies of *Tuber indicum*, *Tuber pseudoexcavatum*, *Tuber sinoaestivum*, *Tuber huidongense* and *T. melanosporum* (*Ye et al., 2018*; *Roux et al., 2016*). Though the ascocarp of *T. melanosporum* itself may not be rich in *Pseudomonas*, it can somehow acquire *Pseudomonas* and massive other nitrogen cycle-related bacteria (*Antony-Babu et al., 2014*). And though these nitrogen cycle-related bacteria were carried in the ascocarp inoculum and brought into the substrate, we still found some *nirS*-type denitrifying bacteria such as Unclassified Gammaproteobacteria and Unclassified Betaproteobacteria significantly less abundant in ectomycorrhizosphere soils than in rhizosphere soils. These may imply that, if without the bringing-in of these bacteria from the ascocarp to the soils, their relative abundance might be lower. This further

revealed that the presence of truffle mycelia inhibited the colonization of AOB and some *nirS*-type denitrifying bacteria. The *Pseudomonas* such as *Pseudomonas fluorescens* could enhance both the mycorrhization of *T. melanosporum* and the nutrient uptake of the host plant (*Dominguez et al., 2012*; *Citterio et al., 2001*; *Mamoun & Olivier, 1992*). The relative abundance of *Pseudomonas* detected in ectomycorrhizosphere soils were significantly greater than that in rhizosphere soils ($P < 0.05$), and the previous study also indicated *Pseudomonas* had greater representation inside the brûlés compared with the outside (*Mello et al., 2013*). We therefore hypothesized that *T. melanosporum* may improve the growth of the host plants by enhancing the colonization of *Pseudomonas* in rhizosphere soils of the host plants in the initial symbiotic stage. The ectomycorrhizosphere soils of the host plant contained a large number of *Pseudomonas* species whose culture condition was unclear and needs further study. We also found that the abundance of *nirS*-type denitrifying bacteria was much greater than that of AOB in the rhizosphere soils of *C. illinoinensis*, and there were a large number of unclassified nitrogen cycling bacteria in ectomycorrhizosphere soils. The dynamic evolution of other nitrogen cycling microorganism including fungal and archaea communities over the process of ectomycorrhizal formation require further exploration.

On account of sterilized seeds and substrates used in our study, bacteria in tap water and air could then randomly get into the soil, and those who get early entrance may be more competitive and dominant due to their earlier starts and more rapid reproduction. This may cause the levels of specific AOB and *nirS*-type denitrifying bacteria which were detected in different samples of the same treatment vary greatly. In fact, we did find that the number of certain OTUs differed greatly in 4 different samples in the same treatment. Therefore, the results obtained when the sample size was small ($n = 4$) may not comprehensively reflect the true situation of all plants under the same culture conditions. It is difficult to guarantee that the significant differences in bacterial communities between two treatments were not accidental. We will set a larger number of samples in a treatment, while by measuring the rate of nitrification and denitrification, the mycorrhizal degree of *T. melanosporum* mycorrhizal synthesis, and the abundance of the dominant AOB and *nirS*-type denitrifying bacteria (especially *Nitrosococcus* and *Pseudomonas*, whose abundances differed greatly in ectomycorrhizosphere and rhizosphere), we could conduct some in-depth researches. The subsequent work could also focus on analyzing the correlation of the above three, and exploring whether the dominant bacteria related to the nitrogen cycle at the initial stage of inoculation could affect the mycorrhizal synthesis by changing the content of available nitrogen in the soil, in addition, isolating and culturing these unclassified bacteria and learning their biological characteristics.

## CONCLUSIONS

In the current study, we found *nirS*-type denitrifying bacterial and AOB communities in *C. illinoinensis* rhizosphere soils were significantly affected by *T. melanosporum* inoculation at the early symbiotic stage, though the total nitrogen, nitrate-nitrogen and ammonium-nitrogen contents of ectomycorrhizosphere soils were not different from

those of rhizosphere soils. There was a lower-abundance of AOB and greater-abundance of *nirS*-type denitrifying bacterial communities in *C. illinoinensis-T. melanosporum* ectomycorrhizosphere soils compared with control treatment. Proteobacteria was the dominant bacterial phylum, and the abundance of *Nitrosomonas* (AOB) was significantly reduced because of *T. melanosporum* inoculation, while the abundance of *Pseudomonas* (*nirS*-type denitrifying bacteria) increased. This work furthers our understanding of rhizosphere microecology and the cultivation of *T. melanosporum-C. illinoinensis*.

### Funding
This work was supported by the National Natural Science Funds of China (No. 31900079). The funders had no role in study design, data collection and analysis, decision to publish, or preparation of the manuscript.

### Grant Disclosures
The following grant information was disclosed by the authors:
National Natural Science Funds of China: 31900079.

### Competing Interests
The authors declare there are no competing interests.

### Author Contributions
- Zongjing Kang performed the experiments, analyzed the data, prepared figures and/or tables, authored or reviewed drafts of the paper, and approved the final draft.
- Jie Zou conceived and designed the experiments, performed the experiments, analyzed the data, prepared figures and/or tables, authored or reviewed drafts of the paper, and approved the final draft.
- Yue Huang analyzed the data, prepared figures and/or tables, authored or reviewed drafts of the paper, checking the spelling mistakes of the manuscript, and approved the final draft.
- Xiaoping Zhang and Bo Zhang performed the experiments, analyzed the data, prepared figures and/or tables, and approved the final draft.
- Lei Ye performed the experiments, authored or reviewed drafts of the paper, and approved the final draft.
- Xiaoping Zhang conceived and designed the experiments, prepared figures and/or tables, and approved the final draft.
- Xiaolin Li conceived and designed the experiments, authored or reviewed drafts of the paper, and approved the final draft.

### DNA Deposition
The following information was supplied regarding the deposition of DNA sequences:
All raw data are available in Sequence Read Archive (SRA): amoA gene sequences (SRX7207409), nirS gene sequences (SRX720739).

## Data Availability

The data is available at NCBI GEO: SRP232888.

## Supplemental Information

Supplemental information for this article can be found online at http://dx.doi.org/10.7717/peerj.9457#supplemental-information.

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
