# Peer review of "Tuber melanosporum shapes nirS-type denitrifying and ammonia-oxidizing bacterial communities in Carya illinoinensis ectomycorrhizosphere soils"

_PeerJ, doi:10.7717/peerj.9457_

## Round 0.1 · original submission · Major Revisions

Thank you for your submission. All of the reviewers have identified a number of very significant concerns regarding study design, raw data and metadata availability, and manuscript presentation that preclude publication of the manuscript in its current form.

Reviewer 1 ·

Basic reporting

There are several methodological problems and the English needs to be completely revised (there are several sentences that are not readable and there are several verb tense errors).
Licterature references are updated.
Raw data are not shared. The gene sequence data are not deposited in a public database

Experimental design

This work compares the nirS-type denitrifying bacterial and ammonia –oxidizing bacterial communities in the soil of pecans inoculated with Tuber melanosporum and in the soil of not inoculated plants in greenhouse conditions.
Several works have been carried out on the microbial communities in truffle mycorrhizosphere in recent years. Truffle bacterial communities were mostly investigated using the 16S rRNA gene as marker. In my opinion this research using the nitrite reductase gene nirS and the ammonia –oxidizing gene as molecular markers, does not add new information. Morover it is not clear why the Authors decided to use these functional gene markers to analyse the bacterial communities in truffle pots.

There are several methodological problems. For example:
Line 117 -119 It is not clear which was the substrate in the pots, and how it was disinfected.
Line 122-125 why were some plants contaminated after 3 months?
Line 128-130. How can you obtain a spore powder with soaked truffles?
Line 133- line 138 How many plants do you consider for treatment thirty or four? The experimental design is not clear

Validity of the findings

The experiments do not reflect the natural conditions (of the field) and thus the results are not interpretable. The soil was disinfected (usually the soil is steam sterilized for the production of truffle mycorrhized plants in order to avoid contaminations) so the bacteria in the uninoculated plants should be the consequence of greenhouse contaminations due to the operators. In contrast, the plants inoculated with T. melanosporum received also the bacterial communities living inside the ascomata.

The authors do not try to interpret the biological significance of the results

Additional comments

This work could be considered only after having completely rewritten it. I am not convinced about the experimental design.
The authors should be supported by a native English speaker with mycological experience in writing the maunuscript.

Here are some examples of scientific problems in the introduction probably due to the bad English:
Line 42- truffles are not one of the many species of Tuber! Truffles are all the species inside the genus Tuber and, for same Authors, all the hypogeous Ascomycetes.
Line 77-78 dentrifing bacteria, which are widely distributed in microbial population such as bacteria, archea……?

·

Basic reporting

There are a number of mistakes and inconsistencies in the basic reporting of the work that need to be corrected before publishing:
- first of all, the names of the figures to which the authors refer in the text is wrong almost all over the manuscript. This makes its reviewing very tedious. For instance Fig 1 is fig 2. Fig 2 does not allow to see the average number of OTU -> this is in Table1; Fig3 is Fig4... In addition, the legend of some supplemental figures (e.g. FigS2) are missing. Please make appropriate corrections; this should have been carefully checked before submitting the manuscript for reviewing.
- secondly, raw data have been submitted to a database and made publicly available but none of the sample names are the same between the paper and the database so it is impossible to use the data that as been shared. This is make the sharing useless and is not ethical; authors must provide the correspondance between the sample name they use in the article and the one they use in their submission to SRA.
- thirdly a number of references are inappropriate while many references to the field and the work that has already been done is missing (see comments below).
- finally the aims that are given at the end of the introduction are not exactly the ones that are tackled in the work; this should be modified (see comments below)

Experimental design

There are a number of information that are missing or need to be clarified in the methods used:
- the number of replicates used is unclear: the authors mentioned that each treament had thirty replicates (l133) but at the end it seems that only four were used for mycorrhization with truffle spores (l137). Can you clarify what are the 30 replicates then ?
- the authors must describe how they verified that mycorrhiza obtained belonged to T. melanosporum. The fact that a spore inoculum of T. Melanosporum was used to inoculate the seedlings do not guarantee that mycorrhiza obtained are from this fungus. Many contaminations can occur.
- Information are missing for amplicon sequencing data analyses: at which level were the data rarefied, statistics were performed as p values are given but we have no information about which test were used, whether data were transformed or not... please add these essential information.

Validity of the findings

There are a number of missing data that do not allow the reader to assess the validity of the results:
1. Authors must provide the sequencing results; how many reads obtained after quality check and processing, how many OTUs...
2. Only a little part of the results is supported by (undescribed/unknown) statistics. All comparisons done between rhizosphere and ectomycorhizosphere must be supported by stats.
3. Steepness of rank abundance curve is not a quantitative parameter and is difficult to evaluate from the graphs provided. What do you mean by sample CK.2 was the most abundant ? Do you mean "Had the highest richness" ? If so why using a rank abundance graph, this does not provide any information about richness but the distribution of the relative abundance of different OTUs. Alpha diversity indices are more appropriate to answer this question.
4. What is the scale in Figure 5 ? It goes from -0.5 up to 4 while data described in the text are relative abundances going from 0 to 100%; none of the value matches between text and graph in figure 5. In addition, was the ordering of samples in the X axis in figure 5 organised by proximity of pattern with other samples (hierarchical clustering or similar method); if so, it demonstrate that there are a lot oh heterogeneity in the data and there very few difference between control samples and mycorrhiza samples. We can expect that none of the differences described between the most abundant OTUs are supported by statistical tests. Anyway, a barplot would be more convincing than a heatmap since differences are quite low and there are not that many OTUs to display.
5. Table 2. Please provide +/- SE and order the genera so that we can compare easily genera between themselves i.e. order by the most to the less abundant in control but not in mycorrhiza. By the way, "environment samples" is not a taxonomic rank; replace by unknown bacteria (if you are sure they are bacteria)
Last. It would be nice to know the level of mycorrhization by T. melano of the seedlings. Indeed, the importance of the finding highly depends on the level of mycorrhization by Tuber. If only a few percent of the roots have these mycorrhiza, the impact on the nitrogen cycle will be low.

Additional comments

This study is innovative and interesting to the field as there is little information about the nitrogen cycle in truffle orchards. The study does not allow to determine whether the differences measured in the relative abundance of AOB and denitrifying bacteria in the ectomycorrhizosphere vs rhizosphere has a significant impact on the nitrogen cycle. Measurements of activities in soil and nitrogen cycles intermediate would have been necessary to be able to truly conclude but it is an interesting first step. However there are a number of points that need to be corrected before publication (in addition to those describe above).

Details comments:
l23: "rarest". T. melanosporum is not the rarest truffle among the one that are commercialized
l42-43 edible fungi which can form -> remove "CAN"
l44: Gogen et al 2012; this ref is not the most appropriate to illustrate this point as it is focused on T. aestivum
l48: costing 1500 euros/kg -> reaching up to instead. Depending on the time of the year and the years, it can drop easily below 1000 €/kg
l49. T.melno contains amino acids and trace elements -> remove; all living organisms contain amino acids and trace elements that's not a specificity of truffles, those are the basal compounds of life on Earth.
l 55: study on mycorrhizal synthesis: what do you mean by mycorrhizal synthesis ? controlled inoculation ? mechanisms of the formation of ectomycorrhiza ?
l76-78: sentence is weird; consider reformulating; not sure to understand the point you want to make saying that dentrifying bacteria are widely distributed among microbial population that include bacteria, fungi and archea.
l85: What do you mean by "bacteria of similar molecular size" ? Relative conservative morphology ? Pseudomonas are rod bacteria
l93-94: few studies have focused ... that's true for the ectomycorrhizosphere of Tuber but not for plants in general. Look for instance ate Henry at al 2008, Environmental Microbiology, many other studies have followed this one
l94: ammoxidation is not the appropriate wording, use ammonium oxidation instead
l105 Esmaeili 2017-> inappropriate citation
l105 the connection between the idea that it is not clear how microorganisms interact with each other and the plant and the fact that you infected C. illinoensis with T. melano to observe mycorrhiza formation is not crystal clear. You may want to rework the connections between ideas in this section.
l108-109: at the same time, we studied.... of microbial communities. This is not true; you did not look at the overall microbial communities but only the one's that have NirS or AmoA genes; this is quite different
l116: "prepared matrix": give details about what it is
L 119-120: The greenhouse... verb is missing
l121: moistened with sterile water -> to reach which level of humidity ?
l123: contamination -> control to verify the absence of ECM contamination ?
Please provide a description of the soil used; what is the content in nitrogen ?
l151 and further: please provide refs for the primers used
l179: "samples were purified" what do you mean ? remove chimera ? contaminants ? how ?
L205: How can ACE1 and CHao index statistically different but not the richness as both are calculated on richness; the variability found in the richness data should be at least as high in CHao and ACE indxes
Lin 212 if not significant then p>0.05 not < same line 238
Fig 4 -the phylum level is not appropriate to compare the data, work at a lower taxonomic level

Discussion section
Authors should compare their results to what has already been published concerning denitrification and ammonium oxidation in other tree rhizospheres (see for instance Moreau et al 2019 Functional Ecology)
Compare also their data with what has been previously described about the microbial communities of ectomycorrhiza of T. melanosporum (Deveau et al. 2016 Mycorrhiza)
line 274: different from what ?
l279: typo in Magnetospirillum
l290: There have been a number of publication that have been made on this topic since Barbieri 2007 (which is not about T.melano by the way but T. magnatum). Similarly there is a number of articles about Pseudomonas in truffles and particularly about T. melanosporum; please expend your literature search

Reviewer 3 ·

Basic reporting

The research presented data on nirS-type denitrifying and ammonia-oxidizing bacterial populations in rhizosphere soils of pecan trees with Tuber melanosporum inoculated roots compared to uninoculated roots. It is interesting, although there are some general issues with the paper.

In particular, the hypotheses are not well defined, and it is not clear what the expectations were. It would have been very nice to measure ecosystem processes (or rates or ammonification / denitrification), for example, to determine how measured communities related to these functions.

Overall, I would recommend improvements on the writing throughout. There are spelling inconsistencies, grammar issues, and details of the methods that need could be improved. I have tried to make note of many of these in the reviewed PDF (attached).

The figures are a bit crude, not clear to read or interpret (particularly Fig 3 & 6).

This paper from 2017 may be relevant to cite:

Mycorrhization of pecan (Carya illinoinensis) with black truffles: Tuber melanosporum and Tuber brumale would be appropriate to cite:
Mycorrhiza (2017) 27:303–309 DOI 10.1007/s00572-016-0743-y

Experimental design

The experiment is kind of minimal but there are 4 replicates of the treatment and 4 replicates of the control (so 8 samples in total).

Having some metrics for comparing control vs. treatment plants and soils would have been beneficial to include (Tuber DNA in soil; % mycorrhizas; hight or biomass of plants, denitrification / ammonification rates, etc.).

The methods are a bit sparse in places. For example, it was hard for me to follow how the nirS and ammonia-oxidizing amplicons were analyzed. That is, how were sequences filtered and clusters determined and identified? What criteria were used to determine species, or was the default 97% OTU cluster definition that is used for 16S rDNA applied to these 2 functional genes? How was taxonomy assigned to the clusters? I am not sure how complete the reference database is. It would have been nice to include 16S rDNA as a comparison for bacteria communities in total. It would also be good to include the sequence output results, total number of reads per sample / per locus post filtering.

Validity of the findings

It appears that Tuber melanosporum ectomycorrhizas lead to a change in bacterial communities (at least nir-s and AMO ones), but only these functional genes were assessed in the community.

Given some of the limitations, it is a bit challenging for me to access how novel this study is, or the implications of the results.

Additional comments

I have included a scan of my edits. I hope they are helpful.

Annotated reviews are not available for download in order to protect the identity of reviewers who chose to remain anonymous.

---

## Round 0.2 · Minor Revisions

Thank you for your willingness to make the changes suggested previously. Both reviewers commented very favorably on the revised manuscript; however a few additional changes are suggested. Please see their suggestions when making your revisions.

Reviewer 1 ·

Basic reporting

The manuscript was considerably improved. However, the English, above all of the new parts need to be revised. Moreover, in my opinion, before to accept this manuscript for publication the Authors should consider that the most importance source of bacterial diversity in the mycorrhized seedlings (which are cultivated in a sterile soil) is the microbioma of the ascomata used to inoculate spores. This important aspect should be incorporated in the discussion. In the point 8 of my previous review I already have raised this point

Experimental design

no comments

Validity of the findings

The discussion need to be improved as suggeste above.

Reviewer 4 ·

Basic reporting

The article is reporting the results of an experiment that tested the impact of black truffle on denitrifying bacteria and ammonia-oxidizing bacteria community structure. The main finding of the experiment showed that the abundance and diversity of ammonia-oxidizing bacteria decreased as a result of ectomycorrhizal colonization of the truffle, but the nirS-type denitrifying bacteria increased in abundance and diversity at the same time.

The authors supplied sufficient background, method descriptions to fully understand their motivation, significance and experimental design and set-up. The structure of the manuscript follows the requirements. Figures and tables look professional and aid the understanding of the manuscript, and seems all needed for the full story.

The English of the manuscript is clearly improved, but there could be improved a bit further (see attached document with tracking and comments for suggestions).

Experimental design

The manuscript describes a small, original experiment to examine the impact of ectomycorrhiza of truffles on nitrogen transferring bacteria in the rhizosphere, and this research is within the aims and scope of the journal.

The authors of the manuscript responded to previous reviewers' comments quite well and made significant improvements to the presentation of their research questions, methods, and explanations. There were a few responses from the authors, which made complete sense in the rebuttal letter, but their comments there did not translate into the main text. I placed comments in the attached text or suggested rewording, where I was still unclear about a few things.

I think that the methods are now described with sufficient details and it could be easily replicated.

Validity of the findings

Although the sample size was small (four replicates per treatment), the limited statistical analysis seems sound and all data analysis is free of speculations. The authors point out some of the limitations and what further studies are needed to support the outcome of this experiment.

The conclusions need a bit more clarifications with rephrasing the statement (I put a note on it in the attached text with a suggestion).

Additional comments

The manuscript significantly improved by taking the previous reviews seriously and making changes accordingly. Language editing could improve the text further. I tried my best to make suggestions in an attached file. Please consider my few comments to make the manuscript even more clear. My comments are directly inserted into the attached file.

Annotated reviews are not available for download in order to protect the identity of reviewers who chose to remain anonymous.

---

## Round 0.3 · Minor Revisions

Thank you for your revised manuscript and for your attention to the reviewer's suggestions. Upon re-reading the manuscript I feel that additional minor revisions are warranted to more clearly address the limited number of samples in this study and the resulting limitations of your conclusions. I also encourage you to think critically about what the variation among samples indicates (i.e., as demonstrated by bar plots some samples do not appear to differ between treatments) and how future work could further elucidate such interactions. In addition, information missing on the statistical tests (i.e., the test statistic and exact P-value for each t-test in Tables 2, 3, and 4; what do the shaded circles in Fig. 6 represent?). Lastly, although the raw data have been deposited in the SRA, I highly encourage you to submit the OTU rep sequences, OTU tables, table of taxonomic assignments per OTU, richness data per sample to an online repository such as GitHub or FigShare such that these data are more transparent and reproducible. Without this information is is nearly impossible to evaluate the quality of the data and robustness of the results.

---

## Round 0.4 · accepted · Accept

Thank you for making the suggested changes to the manuscript. I believe the paragraph that was added to the Discussion addresses the limitations of your study; however, I encourage you to re-check the grammar ahead of publication.